# Controlling the first wave of the COVID–19 pandemic in Malawi: Results from a multi-round study

Jethro Banda[1], Albert N. Dube[1,2], Sarah Brumfield[3], Amelia C. Crampin[1,4,5], Georges Reniers[4], Abena S. Amoah[1,4], Stéphane Helleringer[6]*

1 Malawi Epidemiology and Intervention Research Unit, Lilongwe, Malawi, 2 Department of Community Health, Kamuzu University of Health Sciences, Blantyre, Malawi, 3 Department of Epidemiology, School of Public Health, Boston University, Boston, Massachusetts, United States of America, 4 Department of Population Health, London School of Hygiene and Tropical Medicine, London, United Kingdom, 5 School of Health and Wellbeing, University of Glasgow, Glasgow, Scotland, 6 Program in Social Research and Public Policy, Division of Social Science, New York University-Abu Dhabi, Abu Dhabi, United Arab Emirates

* sh199@nyu.edu

**Data Availability Statement:** All relevant data are within the article and its Supporting information files.

## Abstract

We investigated behavioral responses to COVID–19 in Malawi, where a first wave of the pandemic occurred between June and August 2020. Contrary to many countries on the African continent, the Government of Malawi did not impose a lockdown or a stay-at-home order in response to the initial spread of SARS-CoV-2. We hypothesized that, in the absence of such requirements to restrict social interactions, individuals would primarily seek to reduce the risk of SARS-CoV-2 transmission during contacts, rather than reduce the extent of their social contacts. We analyzed 4 rounds of a panel survey spanning time periods before, during and after the first wave of the COVID-19 pandemic in Malawi. Five hundred and forty-three participants completed 4 survey interviews between April and November 2020. We found that the likelihood of attending various places and events where individuals work and/or socialize remained largely unchanged during that time. Over the same time frame, however, participants reported adopting on a large scale several behaviors that reduce the transmissibility of SARS-CoV-2 during contacts. The percentage of panel participants who reported practicing physical distancing thus increased from 9.8% to 47.0% in rural areas between April-May 2020 and June-July 2020, and from 11.4% to 59.4% in urban areas. The percentage of respondents who reported wearing a facial mask to prevent the spread of SARS-CoV-2 also increased, reaching 67.7% among rural residents in August-September 2020, and 89.6% among urban residents. The pace at which these behaviors were adopted varied between population groups, with early adopters of mask use more commonly found among more educated office workers, residing in urban areas. The adoption of mask use was also initially slower among women, but later caught up with mask use among men. These findings stress the importance of behavioral changes in containing future SARS-CoV-2 outbreaks in settings where access to vaccination remains low. They also highlight the need for targeted outreach to members of socioeconomic groups in which the adoption of protective behaviors, such as mask use, might be delayed.

**Funding:** This study was supported by grant R01HD088516 from the National Institutes of Health (PI: SH). The funders had no role in study design, data collection and analysis, decision to publish, or preparation of the manuscript.

**Competing interests:** The authors have declared that no competing interests exist.

## Introduction

Most African countries experienced a first wave of the COVID-19 pandemic between May and August of 2020 [1]. Multiple surveys of the prevalence of antibodies against SARS-CoV-2 suggest that this initial spread of the novel coronavirus throughout the African continent was likely larger than indicated by routinely reported data on confirmed COVID-19 cases [2–6]. How this pandemic wave was brought under control in resource-limited settings, and in the absence of vaccines, remains however partially understood. In this paper, we describe behavioral responses to the first wave of the COVID-19 pandemic in Malawi, a low-income country in southeastern Africa. We contribute to a better understanding of COVID-19 prevention and control in resource-limited settings in several ways.

First, we uncover how individuals navigated the threat of infection in a country where individuals were only encouraged to stay at home in response to the first wave of the COVID-19 pandemic [7, 8]. By contrast, in many other African countries, early policy responses to the pandemic frequently required (at least some) individuals to stay at home, with limited exceptions [9]. A large stream of the literature on behavioral changes in African countries during the course of the COVID-19 pandemic has thus focused on how individuals have coped with "life under lockdown" [10]. Our work provides a contrasting perspective on behavioral responses to the new health threat posed by COVID-19 in settings with fewer restrictions.

Second, we add to the literature on the prevention of COVID-19 in African countries by analyzing a multi-round data set, in which each participant was interviewed several times. Such longitudinal studies conducted since the emergence of SARS-CoV-2 in African countries have primarily measured the impact of the pandemic on socioeconomic outcomes, including for example food availability, consumption patterns, schooling or labor force participation [11–15]. Studies of the adoption of behaviors protective against the spread of COVID-19 have been predominantly cross-sectional, and have focused on the first few weeks following the detection of SARS-CoV-2 in a country or population [16–19]. Longitudinal studies of behavioral change have been conducted, for example, in Senegal and in South Africa [20–23]. However, some of these studies begun only after SARS-CoV-2 had started spreading widely. They thus provide only a partial perspective on how the first wave of the COVID-19 pandemic was controlled in their communities of interest.

We collected 4 rounds of longitudinal data spanning the first 6 months of the pandemic in Malawi. Our first round of data collection (April-May 2020) started less than 3 weeks after the first case of COVID-19 was detected in the country, and well before the incidence of SARS-CoV-2 began to rise sharply [24]. Subsequent rounds of data collection coincided with key epidemic phases: we interviewed panel participants during a period of rapid epidemic growth (June-July 2020), and during a time of sharp decline in the recorded number of confirmed COVID-19 cases (August-September 2020). Our fourth round of data collection took place when the recorded transmission of SARS-CoV-2 had returned to very low levels (October-November 2020).

We used these longitudinal data to test the following hypothesis:

*In the absence of a lockdown, individuals primarily respond to the health threat posed by COVID-19 by adopting behaviors that reduce the risk of transmission during contacts, rather than by limiting contacts.*

We found support for this hypothesis in our dataset. Over time, study participants reported relying primarily on physical distancing and mask use to prevent the spread of SARS-CoV-2, i.e., practices that are recommended in order to reduce the rate at which SARS-CoV-2 is

transmitted during contacts [25]. Our longitudinal data suggest a temporal correlation between the large-scale adoption of these two preventive practices in our sample and decline in the reported incidence of SARS-CoV-2 in the country.

Finally, we also explored socioeconomic differences in the adoption of these protective behaviors during the course of the first COVID-19 wave in the country. Whereas prior studies have investigated behavioral responses among small and homogenous socioeconomic groups, for example healthcare workers [26], the urban poor in selected cities [16, 27], or residents of small rural areas [19, 28], our study sample was dispersed throughout Malawi. It included residents of urban and rural areas, who engaged in a wide range of economic activities, from agricultural work to tertiary occupations. We used the diversity of this sample, in conjunction with insights from theories of the diffusion of innovations [29], to identify the early adopters and laggards in the adoption of protective behaviors against the spread of SARS-CoV-2 [30].

## Theoretical background

### Epidemiological framework

COVID-19 is an infectious disease, which is caused by the SARS-CoV-2 virus. Some infections with SARS-CoV-2 remain asymptomatic [31], but COVID-19 cases often involve mild or moderate symptoms [32], such as fever, cough and tiredness, as well as a sore throat and a possible loss of smell and taste. In some cases, COVID-19 symptoms can be severe, including difficulty breathing, chest pain and/or confusion [33]. Older people and those with co-morbidities (e.g., diabetes) are at a higher risk of severe COVID-19 illness and death [34]. Some COVID-19 patients with severe respiratory symptoms might require mechanical ventilation [35]. Variations in the case fatality rate of COVID-19 between countries have been associated with the availability of more or less invasive mechanical ventilation devices [36].

The potential for SARS-CoV-2 to spread within a population depends on interactions between features of the virus, the characteristics and behaviors of its (human) hosts and the environment within which these interactions occur. This is summarized by the basic reproduction number, $R_0$, defined as the number of new infections produced by a single infection in a completely susceptible population [37]. An epidemic has the potential to expand when $R_0$ is greater than 1, and it will likely fade away when $R_0$ is below 1. $R_0$ is often expressed as the product of (i) the probability that the pathogen is transmitted when a susceptible individual comes into contact with someone who is infected (transmissibility, noted $\beta$); (ii) the average rate at which susceptible and infected individuals come into contact (noted $\bar{c}$), and (iii) the amount of time (noted $d$) that infected individuals remain contagious [38]. We thus have:

$$R_0 = \beta \cdot \bar{c} \cdot d \tag{1}$$

Controlling SARS-CoV-2 requires reducing one or several of those parameters, so that $R_0$ drops below 1 in a sustained manner.

SARS-CoV-2 can be transmitted before the appearance, or in the absence, of symptoms [31, 39]. Because there are limited effective and affordable clinical options to reduce the duration of illness [40], particularly in resource-limited settings [41], few control strategies target the parameter $d$. Instead, most control strategies focus on modifying the $\beta$ and/or $\bar{c}$ parameters.

Vaccines can be an important tool to reduce $\beta$, as in the case of Measles [42]. However, whereas vaccines against SARS-CoV-2 are highly effective at preventing severe COVID-19 cases, hospitalizations and deaths [43], they have more limited efficacy against transmission of the virus [44, 45]. Vaccines also only became available in late 2020, primarily in high-income countries [46]. In African countries, the roll-out of vaccines did not start until 2021, on a very

limited scale [47]. More than a year and a half after the global vaccine roll-out began, fewer than 50% of adults had been vaccinated in the large majority of African countries [48].

The $\beta$ parameter (i.e., the transmissibility of SARS-CoV-2) can also be reduced through the adoption and sustained implementation of behaviors that limit exposure to viral particles emitted by infected individuals during contacts. Potential behaviors that reduce $\beta$ include increasing hand washing to prevent the transmission of SARS-CoV-2 through fomites [49], sufficient physical distancing to ensure that susceptible individuals cannot be reached by large infectious droplets [50], and wearing facial masks to reduce the risk of infection from droplets and other aerosols [51].

Several environmental factors have been associated with mortality, increased severity of COVID-19 symptoms, and heightened transmissibility of the SARS-CoV-2 virus [52, 53]. These factors include air pollution, wind levels, temperature levels and humidity, for example [54–61]. In indoor spaces, limited air circulation has been linked to an increased risk of transmission of SARS-CoV-2 [62]. Environmental modifications that potentially reduce $\beta$ include installation of air filtration devices, as well as other interventions that increase air flow in indoor spaces, and thus limit the concentration of aerosols containing viral particles [63].

The rate of contact between susceptible and infected individuals ($\bar{c}$) is, in part, determined by levels of population density, i.e., the number of people per square kilometer. In denser areas or settings, opportunities for social interactions through which SARS-CoV-2 can be transmitted are likely more frequent [64, 65]. Similarly, international trade and travel might create long-distance networks of interactions, through which the dissemination of SARS-CoV-2 between countries might occur [66].

Reducing $\bar{c}$ requires encouraging or mandating that individuals limit their social interactions. To prevent importations of SARS-CoV-2, some countries have denied entry, or have applied more or less stringent quarantine and other control measures, to travelers seeking to cross their borders. The effectiveness of these restrictions has however varied between countries and over time [67]. After a positive COVID–19 test or a presumptive diagnosis, patients might also be asked to isolate so that they do not transmit the virus to others in their household, community or networks [68]. At that time, they might be encouraged to inform others they have recently come in contact with that they have been exposed to SARS-CoV-2 [69]. This notification process might be conducted directly by the patient, or might be managed by a contact tracing specialist. In addition, new mobile applications have been developed that register contacts and send out alerts [70]. The notification of recent exposures creates opportunities to encourage to individuals who are at high-risk of infections to get tested for SARS-CoV-2, and to quarantine while waiting for test results or for the completion of set number of days.

In some circumstances, health authorities have temporarily restricted or prohibited attendance of certain places or events where individuals socialize and thus possibly transmit SARS-CoV-2. For example, many countries have decided to close schools or offices, and shift to remote learning/working, at various times during the pandemic [71, 72]. Some countries have imposed restrictions on the maximum number of individuals who can participate in a public meeting or event, or who can interact with each other in public or private spaces. Restrictions on social interactions have also commonly been imposed as periods of "lockdown" or as "stay-at-home" or "shelter-in-place" orders [7, 9], during which individuals are only allowed to leave their homes for restricted reasons and at very limited times.

## Policy responses in African countries

In African countries, as in other regions of the world, governments and health stakeholders have promoted a mix of interventions, which aimed to reduce $\beta$ and $\bar{c}$ simultaneously. Early in

the pandemic, the balance of this mix in many African countries leaned towards measures to reduce $\bar{c}$, particularly through lockdowns, stay-at-home orders and other restrictions on mobility and social interactions (Fig 1). Among the 50 African countries documented by the Oxford COVID-19 Government response tracker [9], 40 countries required (at least some of) their citizens to stay at home at some point during the period stretching from the start of the pandemic to November 1st, 2020. Some of these countries (e.g., Zambia, South Africa, Ghana) granted exemptions from lockdown measures for "essential" services and functions. Other countries (e.g., Botswana, Kenya) adopted stricter measures with only limited exceptions. The enforcement of lockdown measures also varied between countries [73–75], with instances of non-compliance with stay-at-home orders, and possibly hostility towards authorities, documented in multiple settings [76, 77].

Over the same period, five countries only recommended staying at home to their citizens, without mandating a lockdown or imposing stay-at-home orders. These included Mali, Malawi, Côte d'Ivoire, Ethiopia, Niger and Mozambique (Fig 1). Five other countries, including Bénin, Cameroon, Burundi and Tanzania did not take any such measures to try and limit the rate at which their citizens came into contact with one another [78].

Much of the research on behavioral change in African countries during the COVID-19 pandemic has focused on settings where requirements to stay at home have been imposed. Several mathematical models have projected the effects of such restrictions on epidemic trajectories [79, 80]. Two time-use surveys in Kenya and South Africa have shown that lockdowns reduced social interactions in target populations and prompted a decline in estimates of the reproduction number [22, 27]. Various surveys have also documented the prevalence of protective behaviors targeting $\beta$ (i.e., the transmissibility of SARS-CoV-2) in settings where individuals had been required to stay at home [27, 81–89]. A study conducted in Ghana and South Africa thus showed how the urban poor responded to periods of lockdown, indicating that strict regulations did not always translate into increased social distancing due to lack of space and limited information [16]. Finally, several studies have highlighted how the adoption of protective measures depends on collective norms and local perceptions of governmental actions taken in response to the pandemic [90, 91].

In the few countries where the population was only encouraged to stay at home (Fig 1), investigations of local responses to the COVID-19 pandemic have been limited. They have focused on small and selective groups, often during the very early stages of the pandemic. For example, in Ethiopia and in Malawi, studies of attitudes and behaviors related to COVID-19 have been conducted among healthcare workers [92], hospital patients [93, 94] and waiters working in restaurants and bars [95]. These behaviors have also been investigated among internally displaced persons in Mali [86], and residents of small areas affected by conflict in Cameroon [96]. In Ethiopia, Mozambique and Malawi, some studies have documented behavioral responses to COVID-19 in selected small towns or rural communities [14, 19, 97–100]. Larger surveys with national samples, on the other hand, have focused on narrower topics such as preferences for different modalities of COVID-19 testing [17].

## Study context

Malawi is thus one the few countries that did not require its citizens to stay at home during (at least part of) the first wave of the COVID-19 pandemic (Fig 1). It is a low-income country located in southeastern Africa. According to the latest census conducted in the country (in 2018), it has a population of approximately 18 million, and a population density of close to 190 persons per square kilometer [101]. At the time of the census, 86% of the population lived in rural areas, whereas 12% of the population resided in the 4 largest cities (Mzuzu, Lilongwe,

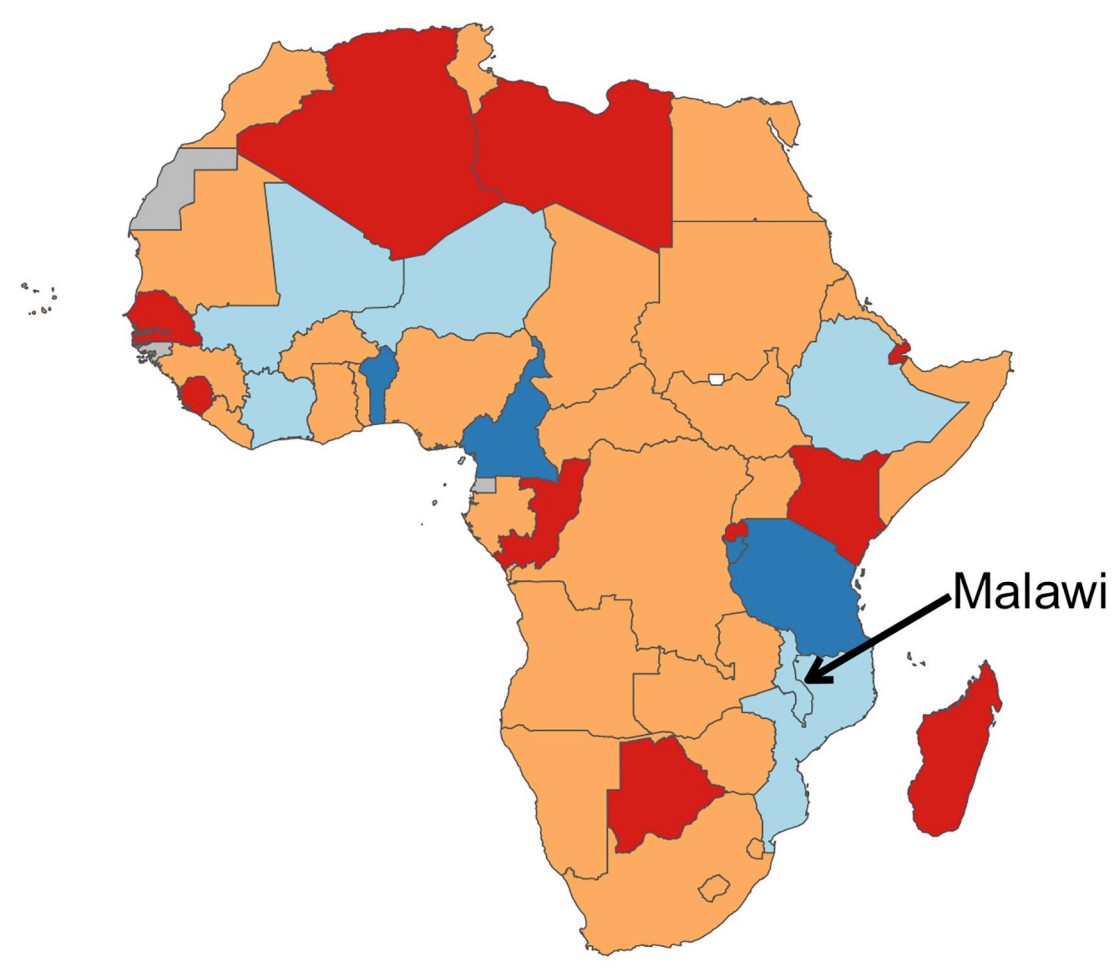

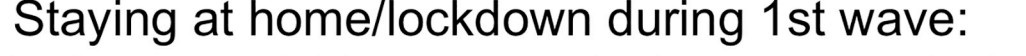

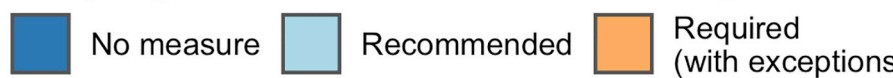

**Fig 1. Adoption of restrictions on social interactions in African countries, from the beginning of the COVID-19 pandemic to October, 1st 2020.**
*Source*: Data on COVID-related policies were extracted from the Oxford COVID-19 Government Response Tracker. Administrative boundaries (Admin 0) were obtained from a World Bank repository of publicly available geographic information (available at: https://datacatalog.worldbank.org/search/dataset/0038272/World-Bank-Official-Boundaries).

Zomba and Blantyre) and the remaining 4 percent in smaller district capitals also classified as "urban" by the national statistical office. Malawi had a GDP per capita of approximately 583 USD and an estimated life expectancy of 63.7 years in 2019 [102].

The population of Malawi is very young, with a median age of 17 years in 2018, and only 4 percent of the total population aged 65 years and older [101]. It is also growing rapidly, at an annual rate of approximately 3 percent per year. Prior to the COVID-19 pandemic, Malawi had made significant progress towards the achievement of health goals related to HIV

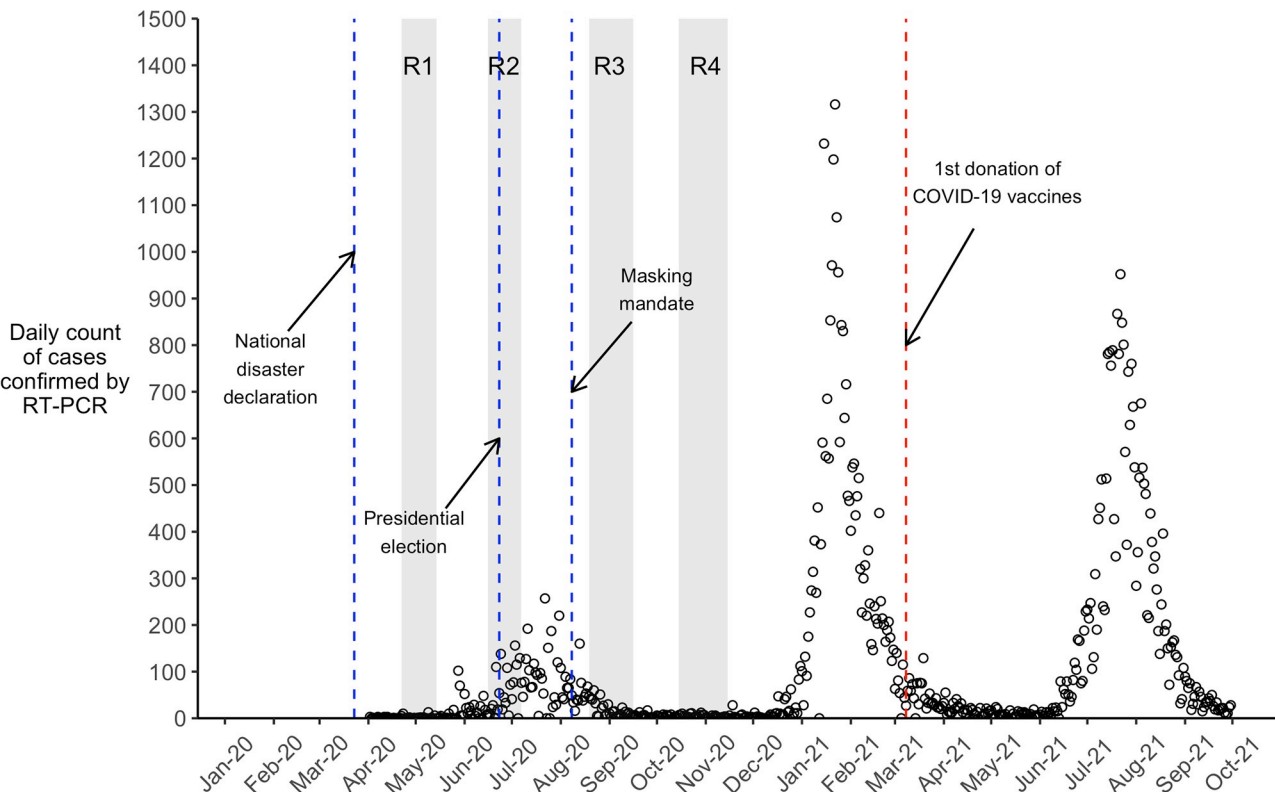

**Fig 2. Recorded COVID-19 cases in Malawi and study timelines.** *Notes*: Data plotted in this graph were drawn from daily reports from the Public Health Institute of Malawi, and obtained via the data portal of the European Center for Disease Control (ECDC). The "o" symbols represent the counts of confirmed COVID-19 cases reported on a given day. The shaded area noted "R1" refers to round 1 of data collection, which occurred between April 22nd and May 14th, 2020. Areas noted "R2", "R3" and "R4" refer to subsequent rounds of data collection. Vertical dashed lines indicate significant events in the course of the pandemic in Malawi.

treatment [103, 104] and to the reduction of child mortality [105]. In recent years, the incidence of non-communicable diseases (e.g., neoplasms) has increased in urban and rural areas [106].

Malawi was one of the last countries to record a case of COVID–19. This occurred on April 2nd 2020, when infection was confirmed in a traveler recently returned to the country (Fig 2). Sporadic clusters of COVID–19 cases were then detected in urban centers for several weeks. In late May, additional importations of COVID–19 cases occurred among migrants returning primarily from South Africa, where SARS-CoV-2 was spreading extensively. The recorded incidence of SARS-CoV-2 increased sharply in Malawi in June and July 2020, before starting to decline in August 2020. After this first wave of the COVID-19 pandemic in the country, the daily number of recorded new cases remained very low until the last few weeks of 2020, when a 2nd wave of SARS-CoV-2 spread started to unfold in the country (Fig 2).

Early in the pandemic, it was projected that 1 in 5 Malawians might become infected with SARS-CoV-2 in 2020, resulting in more than 60,000 hospitalizations and close to 2,000 deaths [107]. These projections accounted for some of the key factors that might affect the transmission of SARS-CoV-2 in Malawi relative to other African countries, including the large proportion of the population residing in rural areas, and expected levels of rainfall. As of the end of December 2020 however, Malawi had recorded 6,583 cases of COVID–19, resulting in 189 deaths. The districts with the most confirmed cases were those where the main cities are

located (i.e., Lilongwe, Blantyre). Subsequently, Malawi experienced additional waves of the pandemic, which were linked to new variants of the SARS-CoV-2 virus. These occurred in early 2021, with peaks at more than 1,300 recorded confirmed cases in a day, and in July and August 2021, with peaks at approximately 1,000 recorded confirmed cases in a day. Our longitudinal data however do not cover these periods (Fig 2).

The Government of Malawi adopted a series of measures to prevent SARS-CoV-2 spread [8]. Prior to the emergence of the first COVID–19 cases in the country, the Government declared COVID-19 a "national disaster" (Fig 2), reduced entries into the country, and implemented COVID–19 screening at border posts that remained open to allow importation of essential goods. The Ministry of Health (MoH) and other stakeholders, initiated information campaigns (e.g., via radio messages, loudspeakers or SMS) to increase awareness and knowledge of the pandemic among the population. This included communications about how SARS-CoV-2 spreads, information about possible symptoms, and what to do when presenting such symptoms. The MoH established a toll-free hotline, as well as social media pages, where updates about the epidemic situation in Malawi, and recommendations, were posted. The MoH also conducted daily briefings about the epidemic situation, during which a spokesperson provided updated information about new cases of COVID-19 confirmed in the country, and periodically diffused messages about preventive behaviors.

To reduce the transmissibility ($\beta$) of the novel coronavirus, the MoH and other stakeholders have rapidly promoted hand washing and physical distancing. Early in the outbreak, the MoH emphasized the use of facial masks to prevent the transmission of SARS-CoV-2. Medical masks and respirators were recommended in healthcare settings, and among people who care for patients with COVID-19, whereas the general public was advised to use cloth masks "in settings where social distancing is not possible and where there is widespread community transmission" [108]. To respond to the increased spread of SARS-CoV-2 in June and July (see Fig 1), the Government made mask wearing mandatory in public in early August. The measure was accompanied by a fine of 10,000 Malawian Kwachas for non-compliers (i.e., approximately 12 USD, or equivalent to 7 days of average personal income).

As part of the national disaster declaration at the end of March, schools and universities were closed, and attendance of public gatherings was limited to 100 people. A national lockdown was announced on April 18th, but several organizations contested the proposed plan. Following legal procedures, the Malawi High Court suspended the implementation of this "stay-at-home" order. Unlike many other countries in Africa (Fig 1), Malawi thus never experienced a lockdown in 2020. The government of Malawi nonetheless adopted other measures to reduce contacts in specific populations. For example, some prisoners were released in August 2020 to reduce overcrowding, and thus prevent large outbreaks of SARS-CoV-2, in jails. Schools and universities reopened in September 2020. International travel also resumed on a larger scale at that time, before being interrupted again in December in response to the rebound in incidence of SARS-CoV-2 (Fig 2).

## Materials and methods

### Study sample and data

We initiated our longitudinal study in April 2020, roughly a month after the Government of Malawi declared COVID–19 a "national disaster", and less than 3 weeks after the first cases of COVID–19 were detected in Lilongwe, the capital city (Fig 2). This "COVID–19 panel' includes 4 rounds of data collection conducted approximately 6 weeks apart and ending in mid-November 2020. During that time, the recorded incidence of SARS-CoV-2 varied greatly, with peaks of close to 200 recorded cases per day in July (Fig 2). The first two rounds of data

collection took place during a time when campaigning for the presidential election of June 23rd was under way [109].

Respondents in the COVID–19 panel had previously participated in a study of the measurement of adult mortality and health-related risk behaviors [110]. This "pre-COVID–19 study" was nested within the activities of a health and demographic surveillance system (HDSS) located in Karonga district, in northern Malawi [111, 112]. This is an area bordering Lake Malawi to the east and located within 1.5-hour drive of the Tanzania border (S1 Appendix). Its economy is predominantly organized around fishing, subsistence farming and small-scale retail activities.

The last round of interviews for the pre-COVID–19 study occurred between December 2019 and March 2020. At that time, we sought to obtain the mobile phone numbers of 1) a random sample of current residents of several HDSS population clusters located close to the lakeshore community of Chilumba, 2) a random sample of former HDSS residents who had migrated throughout Malawi, and 3) the siblings of these current and former HDSS residents. This latter group was recruited to assess the reliability of survey data on mortality among relatives. When the spread of SARS-CoV-2 accelerated throughout the world (including in parts of eastern and southern Africa) in March 2020, we launched the COVID–19 panel: we contacted all participants of the pre-COVID-19 study for whom a phone number was available, and invited them to participate in a series of COVID-related interviews. Referred siblings were also invited to participate in the COVID-19 panel in order to increase available sample size and allow the investigation of urban/rural differences in behaviors during the COVID-19 pandemic.

In total, we sought to obtain the phone numbers of 1,036 men and women aged 18 years and older, who participated in the pre-COVID–19 study. We were successful for 779 participants (75.2%). Among those, approximately 80% agreed to enroll in the COVID–19 panel (n = 619). More detailed descriptions of the constitution of the COVID–19 panel are available elsewhere [24]. The majority of participants were residents of Karonga district. Other participants in the COVID–19 panel were dispersed throughout the country (S1 Appendix), including in large urban centers such as Mzuzu, Lilongwe, Blantyre and Zomba. Due to its sampling frame and recruitment procedures, the COVID–19 panel is not representative of the populations of Malawi or Karonga district.

All interviews of the COVID–19 panel were conducted by mobile phone, by trained interviewers. As in other phone surveys conducted during the COVID–19 pandemic in African countries [91, 113], data collectors conducted interviews from their own homes. Prior to each interview, interviewers introduced themselves to the respondents as working for a nongovernmental research organization. We sought verbal consent prior to enrolling participants in the COVID-19 panel, and then before each interview. At each round of data collection, respondents who completed the interview were provided with 1,200 Malawian Kwachas worth of mobile phone units (approximately 1.6 US Dollars, equivalent to 1 day of average personal income). The protocol of this study was approved by institutional review boards at Johns Hopkins University (IRB00009766) and the London School of Hygiene and Tropical Medicine (21876/RR/25189), and by the National Health Science Research Committee in Malawi (18/03/1996). Recruitment into the COVID-19 study began on April 22, 2020 and lasted until May 15, 2020. Follow-up rates from one round of the COVID–19 panel to the next were 93–95%, and in total, 543 respondents completed all 4 rounds of the COVID–19 panel.

## Measures and variables

During the first round of data collection, we ascertained the socio-demographic characteristics of participants, including their gender, age, educational level and main occupation. Based on

these data, we grouped participants in broad 10-year age groups, except for the youngest respondents who were classified in an age group starting at age 18 and ending at age 24. Similarly, the oldest respondents were classified in an open-ended interval starting at age 55. We constructed a categorical variable, which classified participants as having completed either no schooling or only primary-level grades, secondary-level grades or university-level and other institutions of higher education. Participants' main occupation was represented using a categorical variable, with levels including office work, business and sales, manual work, farming/fishing and other occupations.

At each round of data collection, we reassessed the location of participants, including their district of residence, and whether their residence was in an urban or rural area. We used the definition of "urban" areas from the Malawi National Statistical Office: "urban centers in Malawi refer to the four major cities [of] Blantyre, Lilongwe, Mzuzu, and Zomba, town councils and bomas (i.e., the administrative capital of each district) and other town-planning areas" [101]. All other areas of residence were classified as "rural".

To explore changes in $\bar{c}$, i.e., the rate of contact between infected and susceptible individuals, we asked respondents how many people resided in their households, and how many rooms their house included. These data were collected during each round of interviews, and we used them to calculate an "occupancy ratio", i.e., the number of household residents divided by the number of rooms. This is an indicator of the potential for intra-household exposure to SARS-CoV-2 and other respiratory pathogens [114, 115]. To measure contacts outside of the household, we drew a list of places where people commonly interact in Malawi, which included markets, churches and mosques, neighbors or relatives' houses, workplaces, football games, baptisms or funerals. During each round of data collection, we then asked respondents if they had attended each of these places/events in the past 7 days before the interview. Based on these data, we created a series of binary variables taking value 1 if the participant reported having attended a specific place/event, and 0 otherwise.

During each interview, we also asked respondents to list the behaviors they had used to reduce the spread of SARS-CoV-2 in the past month. This question was adapted from instruments previously administered in COVID-related surveys in high-income countries [116]. During pre-testing, we constituted a list of possible answers to this question. This list included behaviors that reduce $\beta$, e.g., use of facial masks, physical distancing or handwashing, and behaviors that reduce $\bar{c}$, e.g., avoiding going out in general. It also included some behaviors that might jointly affect $\beta$ and $\bar{c}$, e.g., avoiding crowded areas. Interviewers did not read this list to respondents. They let them spontaneously recall behaviors, and coded their answers using the list of potential behaviors. If a behavior was not included in the list, they coded the answer as "other" and specified the respondent's answer in a follow-up field. Multiple answers were allowed. After each answer, interviewers were instructed to probe in a non-specific manner, by asking respondents if there was anything else that they did to prevent the spread of SARS-CoV-2. Based on the answers elicited from this question, we created a series of binary variables taking value 1 if the participant reported having implemented a specific behavior to reduce the spread of SARS-CoV-2, and 0 if they had not adopted this behavior.

Finally, in rounds 3 (August-September 2020) and 4 (October-November 2020), we asked respondents who reported wearing a facial mask in the past month how often they wore a mask when out in public. Possible answers were "always", "often", "sometimes" and "rarely". Based on these reports, we formed a categorical variable describing consistency of mask use among study participants.

## Data analysis

Our primary goal was to test the hypothesis that, in the absence of a requirement to stay at home, individuals primarily respond to the threat of infection by implementing behaviors that reduce $\beta$, i.e., the transmissibility of SARS-CoV-2, rather than behaviors that reduce $\bar{c}$. To do so, we assessed how reports of behaviors related to $\bar{c}$ and $\beta$ changed between consecutive rounds of data collection. If the prevalence of behaviors that reduce $\beta$ increased during periods of epidemic spread, while the prevalence of behaviors that affect $\bar{c}$ remained stable or even declined, then our hypothesis would be supported. On the other hand, if only behaviors that affect $\bar{c}$ or if both types of behaviors changed over time, then our hypothesis would be contradicted.

To detect such trends, we used the following empirical strategy. We first calculated and visualized the proportions of participants who reported that they had attended a specific place/event, or engaged in a given protective behavior, during the 7-day or 1-month periods prior to a data collection round, respectively. We did so separately for reports of attendances of places and events, for reports of behaviors affecting $\beta$ (e.g., mask use) and for reports of behaviors at least partly affecting $\bar{c}$ (e.g., not leaving the house). In calculating confidence intervals around the proportions of participants who had attended specific places/events, or had engaged in various behaviors, we accounted for the multiple observations per respondent that are included in our panel study. We visualized these data by place of residence because in Malawi, the PCR-confirmed cases were predominantly recorded in cities.

Given that our longitudinal data covers periods of epidemic growth (e.g., June-July 2020), as well as low circulation of the coronavirus (e.g., October-November 2020), we explored potential temporal associations between epidemic dynamics and changes in the proportion of participants who report engaging in a given behavior. Such variations in rounds 2 or 3, for example, would indicate that the accelerated adoption of protective behaviors might be correlated with epidemic growth and/or the promulgation of the mask mandate (Fig 2). Using data from rounds 3 & 4, we also investigated how the consistency of mask use, and the reported reasons for wearing masks varied at a time when the spread of SARS-CoV-2 was receding in the country (Fig 2). All these analyses and visualizations were conducted using the statistical software, R.

We then investigated patterns of adoption of mask use and physical distancing in our sample. These are the two behavioral strategies that became more prevalent during the course of the COVID-19 panel (see below). We classified panel participants into 3 three categories, according to the timing of their adoption of each of those two behaviors. In doing so, we followed groupings derived from theories about the diffusion of innovations [29]. "Innovators/ early adopters" were those who reported mask/physical distancing use in round 1 of the study (April-May), shortly after the MoH formulated recommendations to adopt these behaviors. "Majority adopters" were those who first reported mask use/physical distancing in rounds 2 or 3 (June-July, or August-September). "Laggards" were those who never reported mask use or physical distancing in rounds 1 through 4, or who first reported these behaviors during round 4 (October-November), i.e., after the first wave of the pandemic had subsided.

To better understand how these protective behaviors diffused within our study sample, we measured the socioeconomic correlates of the adoption of mask use and physical distancing. We used ordered logit models to assess the relations between dependent variables describing adoption patterns, and independent variables describing participants' age group, sex, educational level, occupation, and place of residence at baseline. We coded our dependent variables as 1 = laggards, 2 = majority adopters and 3 = early adopters, so that positive model coefficients reflect the likelihood of an earlier adoption of mask use or physical distancing.

Ordered logit models make a proportional odds assumption, i.e., they consider that the association of each independent variable with the outcome does not vary across categories of the outcome [117]. In standard ordered logit models, this assumption must hold for all coefficients. Instead, we used generalized ordered logit models that allow relaxing this assumption for some of our models' coefficients [30]. To select those coefficients, we conducted Wald tests of the proportional odds assumption for each independent variable included in the model(s). Standard errors were adjusted for the clustering of observations within families. To estimate generalized ordered logit models, we used the gologit2 command in Stata 15.1 [118].

## Results

The socio-demographic characteristics of study participants are presented in Table 1. In total, 419 participants resided in rural areas at round 1 (April-May 2020), whereas 124 resided in urban areas. In both settings, approximately 60% of participants were women. The age distribution of participants was similar in urban and rural areas, but their main economic activity and their educational level differed markedly. In rural areas, close to half of the participants reported engaging in agriculture or fishing (193/419, 46.1%), whereas in urban areas, the most common occupations were office work (27/124, 22.1%) and business/sales (35/124, 28.7%). In

**Table 1. Characteristics of panel participants, by area of residence during the first round of data collection, Malawi (April–November 2020).**

| Area of Residence in April-May 2020 | Rural areas | Urban areas |
|---|---|---|
| Gender of the respondent | | |
| Male | 169 (40.3%) | 49 (39.5%) |
| Female | 250 (59.7%) | 75 (60.5%) |
| Age | | |
| 18–24 | 76 (18.1%) | 25 (20.2%) |
| 25–34 | 144 (34.4%) | 42 (33.9%) |
| 35–44 | 133 (31.7%) | 41 (33.1%) |
| 45–54 | 55 (13.1%) | 14 (11.3%) |
| $\geq$55 | 11 (2.6%) | 2 (1.6%) |
| Occupation[1] | | |
| Office work | 17 (4.1%) | 27 (22.1%) |
| Business, sales and services | 104 (24.8%) | 35 (28.7%) |
| Manual work | 55 (13.1%) | 26 (21.3%) |
| Agriculture/fishing[2] | 193 (46.1%) | 9 (7.4%) |
| Other[3] | 5 (11.9%) | 25 (20.5%) |
| Educational level | | |
| None/Primary school | 211 (50.4%) | 23 (18.6%) |
| Secondary school | 186 (44.4%) | 64 (51.6%) |
| Higher education | 22 (5.2%) | 37 (29.8%) |
| N | 419 | 124 |

Notes: all figures in parentheses are column percentages;

[1] All data reported in the table were collected in April-May 2020, except occupation, which was measured during round 2 of the COVID–19 panel in June-July 2020;

[2] Some urban respondents who reported agriculture/fishing as their main activity may have been temporary city residents.

[3] The category "other" includes primarily respondents whose main activity is domestic work, as well as students and retirees.

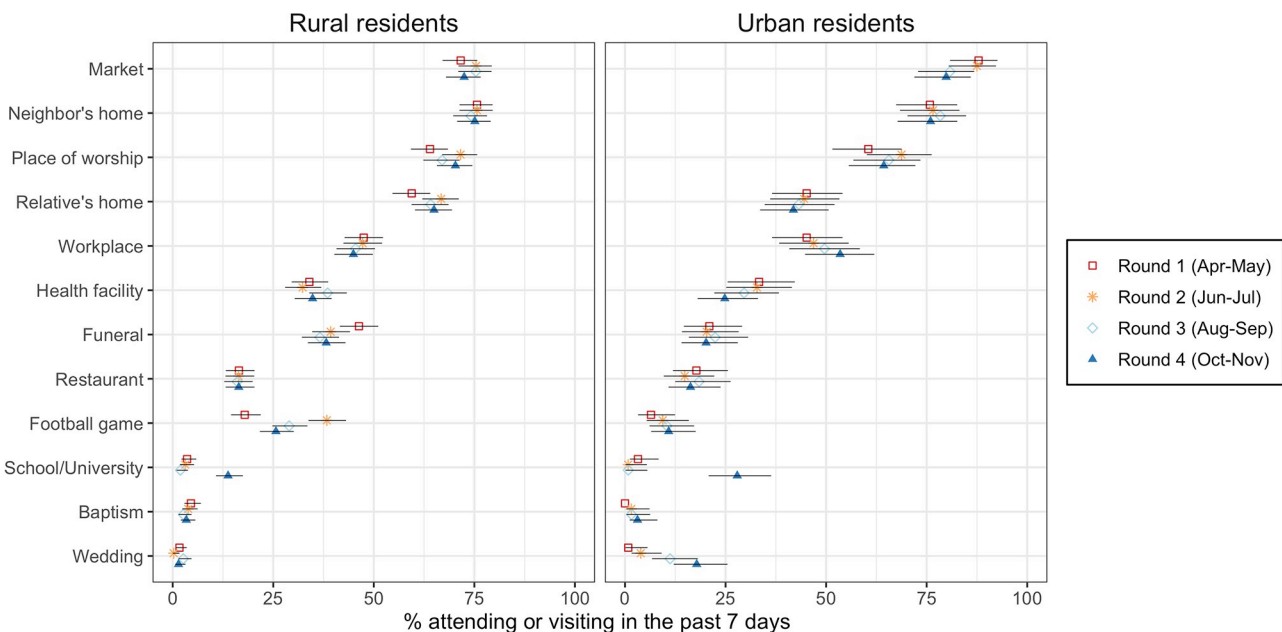

**Fig 3. Attendance of places and events in Malawi (April to November 2020).** *Notes*: the data plotted in this graph were collected from a series of questions asking respondents whether they had attended the places listed on the y-axis in the past 7 days prior to the survey. The places that appear on the y-axis are ordered according to their attendance in urban areas in round 1. Error bars represent 95% confidence intervals. In calculating confidence intervals, standard errors were adjusted for the clustering of observations within respondents.

rural areas, more than half of the participants had never attended school, or had only attended primary grades. In urban areas, more than half of the participants had attended secondary schools, and close to 1 in 3 participants had pursued higher education.

In rural areas, the median household occupancy ratio declined from slightly more than 2 household members per room in round 1 (April-May), to approximately 1.85 household members per room in subsequent rounds (S2 Appendix). In urban areas, there were no changes in household occupancy ratios over time.

There were limited changes in attendance of places and events where people socialize across all 4 rounds of data collection (Fig 3). Rural and urban participants reported most commonly visiting markets, their neighbors' homes, and places of worship. The proportion of urban residents who reported visiting a health facility in the past week declined over time, whereas the proportion of rural participants who reported attending football games varied between rounds.

The most salient changes in attendance of places and events occurred in the 4th round of data collection (October-November 2020), i.e., after the first wave of the pandemic had passed. Attendance of schools/universities thus increased sharply after these institutions reopened in September 2020. For example, whereas <1% of urban residents visited a school or university in round 3 (August-September 2020), this proportion increased to 27.9% in round 4 (October-November 2020). Similarly, the proportion of urban residents who reported attending a wedding in the past week increased from <1% in round 1 (April-May 2020) to 17.8% in round 4 (October-November 2020).

In each study round, <1% of participants reported that they had not done anything to prevent the spread of SARS-CoV-2. In round 1 (April-May 2020), more than two thirds of panel participants in urban areas, and close to half of participants in rural areas, reported "avoiding crowded areas" to reduce the spread of SARS-CoV-2 (S3 Appendix). The proportion of

participants who reported this behavior declined sharply in latter rounds: for example, only one third of urban participants reported avoiding crowded areas in round 2 (June-July 2020). At each study round, approximately one in four participants, regardless of their residence, reported that they avoided "going out in general" to reduce the spread of SARS-CoV-2. Initially, a small number of participants also reported "avoiding visibly sick people" to reduce the spread of SARS-CoV-2 (S3 Appendix), but this behavior was rapidly abandoned. Very few respondents reported reducing their mobility within Malawi to prevent the spread of SARS-CoV-2.

Study participants reported practicing several behaviors that modify the transmissibility of SARS-CoV-2. In each round, more than 90% of participants reported washing their hands more often (Fig 4). In round 1 (April-May), the next most commonly reported behaviors affecting transmissibility were the use of hand sanitizer among urban respondents (22.8%) and covering up coughs/sneezes among rural residents (23.2%). In subsequent rounds, the prevalence of these behaviors either declined or increased only slightly. Instead, panel participants shifted to two other strategies as their main protective behaviors.

Whereas only 9.8% of rural residents, and 11.4% of urban participants, reported that they had practiced physical distancing in the past month during round 1 (April-May 2020), these proportions increased to 47.0% and 59.4% in round 2 (June-July 2020), respectively. In subsequent data collection rounds, the proportions of participants who reported engaging in physical distancing further increased in rural areas, and remained stable in urban areas.

In all data collection rounds, the use of facial masks in the month prior to the survey was more widespread among participants in urban areas than in rural areas (Fig 4). Reported mask use did not increase markedly between rounds 1 and 2 among participants in rural areas (5.6% and 8.4%, respectively), whereas the proportion of urban participants who reported using a mask almost doubled during the same timeframe (from 20.3% to 38.2%). The proportion of

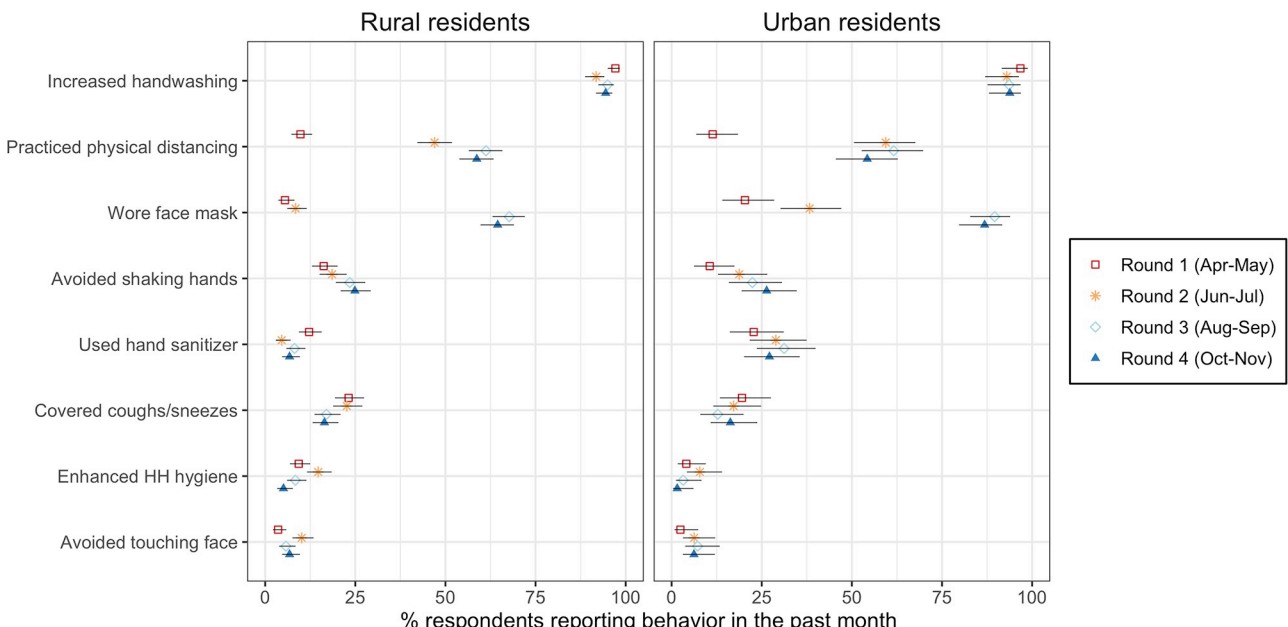

**Fig 4. Adoption of behaviors to reduce transmissibility of SARS-CoV2 in Malawi (April to November 2020).** *Notes*: the data plotted in this graph were collected from a series of questions asking respondents what they had done to prevent the spread of SARS-CoV-2 in the past month. The behaviors that appear on the y-axis are ordered according to their prevalence in urban areas in round 1. Error bars represent 95% confidence intervals. In calculating confidence intervals, standard errors were adjusted for the clustering of observations within respondents.

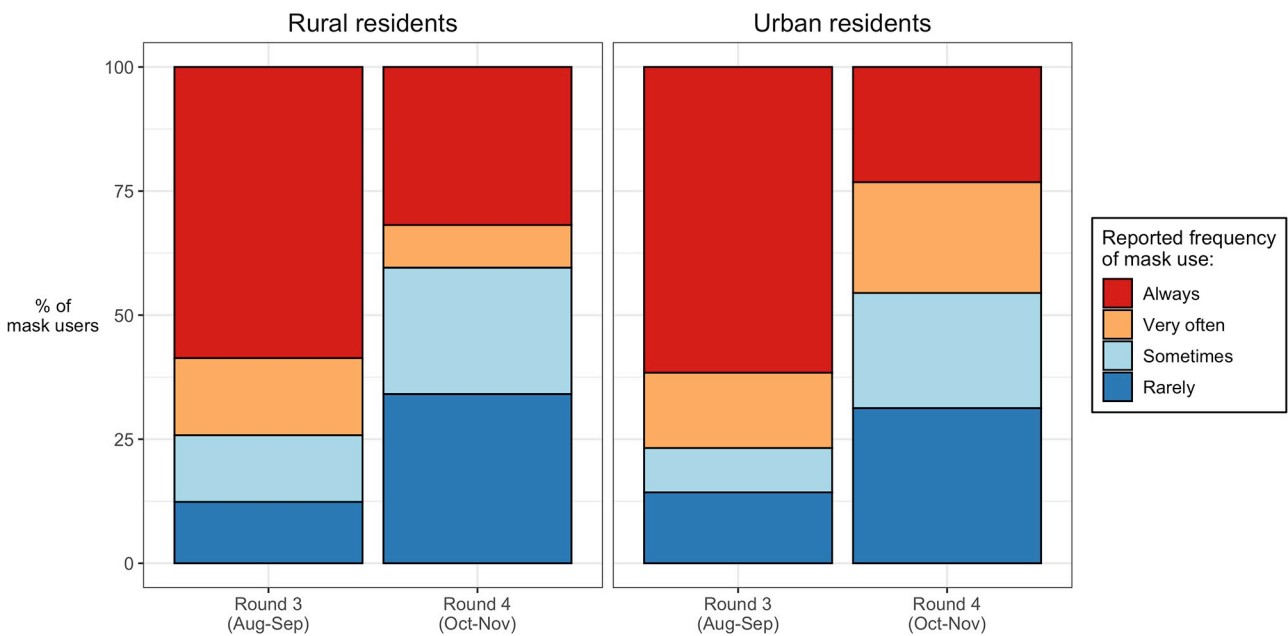

**Fig 5. Frequency of mask use in Malawi (August to November 2020).** *Notes*: in rounds 3 & 4, we asked respondents who reported that they had used a mask to prevent the spread of SARS-CoV-2, how often they had done so. These data were not collected in rounds 1 & 2.

participants reporting use of facial masks then increased sharply in subsequent rounds: in round 3 (August-September 2020), 89.6% of participants in urban areas, and 67.7% in rural areas, reported that they had worn a mask in the past month. These proportions remained largely stable in round 4 (October-November 2020) in all areas, with 64.5% of rural residents, and 86.8% of urban residents, reporting the use of face masks in the past month.

The consistency of mask usage however declined after the transmission of SARS-CoV-2 returned to very low levels in Malawi (Fig 5). In rural areas, the proportions of mask users who reported wearing masks either "always" or "very often" when out in public declined from 71.2% to 39.7% between rounds 3 (August-September) and 4 (October-November). In urban areas, corresponding figures were 78.9% and 44.7%.

Whereas the near-universal adoption of increased hand washing occurred rapidly (Fig 4), wearing facial masks and physical distancing emerged gradually as the other main behavioral strategies to reduce the spread of SARS-CoV-2 among panel participants. In Table 2, we explored the correlates of the adoption of these behaviors. The earlier adoption of mask use was associated with residence in urban areas, and higher levels of education. Compared to participants whose main occupation was in office work, those engaged in business, sales or services, manual work, fishing and farming or other activities (e.g., domestic work, students) were less likely to adopt mask use early in the pandemic. Women were also less likely to be early adopters of mask use than men (coefficient = -1.13, p<0.001) but their adoption of this practice caught up with the men's uptake of masks in rounds 2 (June-July 2020) & 3 (August-September 2020). As a result, the coefficient associated with the likelihood that women were majority/early adopters rather than laggards was 0.10 (not significant).

Our model of the adoption of physical distancing did not uncover any variable associated with adoption patterns. Similar results were obtained with more detailed categorizations of adoption patterns, that split "majority adopters" into "early" and "late" majority adopters.

**Table 2. Generalized ordered logit models of the adoption of protective measures against SARS-CoV-2/COVID-19, Malawi (April–November 2020).**

| Covariates | Adoption of Mask use | Adoption of Physical Distancing |
|---|---|---|
| Gender of the respondent | | |
| Male | Ref | Ref |
| Female | - -[a] | -0.21 (0.22) |
| Female (laggard vs. majority/early) | 0.10 (0.23) | - - |
| Female (laggard/majority vs. early) | -1.13 (0.34)*** | - - |
| Area of residence | | |
| Rural areas | Ref | Ref |
| Urban Areas | 1.17 (0.28)*** | 0.17 (0.23) |
| Age | | |
| 18–24 | Ref | Ref |
| 25–34 | -0.27 (0.28) | -0.23 (0.29) |
| 35–44 | -0.22 (0.28) | -0.22 (0.28) |
| 45–54 | 0.16 (0.35) | 0.40 (0.37) |
| $\geq$55 | -0.09 (0.53) | -0.13 (0.74) |
| Occupation[1] | | |
| Office work | Ref | Ref |
| Business, sales and services | -1.21 (0.41)** | - -[a] |
| Business, sales and services (laggard vs. majority/early) | - - | -0.60 (0.45) |
| Business, sales and services (laggard/majority vs. early) | - - | 0.17 (0.47) |
| Manual work | -0.92 (0.47)* | -0.04 (0.44) |
| Farming and fishing | -1.44 (0.45)*** | -0.06 (0.43) |
| Other | -1.70 (0.47)*** | -0.32 (0.46) |
| Educational level | | |
| None/Primary school | Ref | Ref |
| Secondary school | 0.51 (0.22)* | 0.22 (0.20) |
| University/Higher education | 0.77 (0.36)* | 0.02 (0.35) |
| Constant | | |
| laggard vs. majority/early | 0.72 (0.73) | 1.57 (0.67)* |
| laggard/majority vs. early | -1.43 (0.75) | -2.10 (0.67)** |
| **Model statistics** | | |
| N (observations) | 540 | 540 |
| N (clusters) | 271 | 271 |
| Wald $\chi^2$ | 84.30*** | 18.77 |
| Pseudo-$R^2$ | 0.109 | 0.019 |

**Notes**: Numbers in the table are coefficient estimates, whereas numbers in parentheses are standard errors (adjusted for the clustering of observations in families).

[a] The proportional odds assumption is violated for this covariate, so generalized modesl estimate two coefficients.

*** p<0.001,

** p<0.01,

* p<0.05

## Discussion

We used survey data collected by mobile phone to investigate how individuals navigated the new health threat posed by the COVID-19 pandemic in Malawi, a country where health authorities did not require population members to stay at home in response to the initial spread of SARS-CoV-2 in 2020. We hypothesized that, in such a context, individuals would favor the adoption of behaviors that reduce the transmissibility of SARS-CoV-2 during contacts, rather than modify their patterns of contacts. We found strong support for this hypothesis. Some respondents reported avoiding going out and staying away from crowded areas in the first few weeks of the pandemic in Malawi, but these behaviors did not diffuse among our panel participants. Instead, most patterns of engagement in social events or attendance of places where people interact remained largely stable throughout the course of our panel study.

On the other hand, panel participants reported adopting on a large scale two strategies that reduce the transmissibility of SARS-CoV-2 during contacts: physical distancing and the use of facial masks. The timing of their adoption of these behaviors preceded or coincided with an epidemic downturn, observed in Malawi in late July/early August 2020. Increased mask usage also followed the promulgation of a mandate to wear facial masks when out in public. Our study thus suggests that behaviors that limit the transmissibility of SARS-CoV-2 might have played an important role in controlling the first wave of the COVID-19 pandemic in Malawi.

These results contrast with findings from studies conducted in other African countries, where individuals were required to stay at home for some time, and with often limited exceptions, in order to stem the spread of SARS-CoV-2. In these countries, lockdowns and similar orders to stay-at-home or shelter-in-place significantly altered patterns of social interactions within local populations. By reducing the extent of contacts between infected and susceptible individuals, they likely played a key role in controlling the spread of SARS-CoV-2 during this first wave of the pandemic [10, 22, 27]. Such restrictions, however, had a negative impact on various aspects of social and economic life, threatening the resilience of many households [119]. They might have exacerbated inequalities, for example by limiting opportunities to trade and work for members of the poorest households, as in South Africa [10]. In our study in Malawi, attendance of places where economic activities are conducted (e.g., markets, workplaces) remained stable or even slightly increased during the first 6 months of the COVID-19 pandemic in Malawi (Fig 3). A COVID-19 response that emphasizes reducing the transmissibility of SARS-CoV-2 might thus be less disruptive of local livelihoods than strategies that imposed restrictions on contacts and interactions. To better evaluate the sustainability of these different approaches to COVID-19 prevention in resource-limited settings, further studies are needed that include additional outcomes such as food security.

The role of mask use and physical distancing in limiting the spread of SARS-CoV-2 in the absence of vaccines has also been highlighted in a number of other studies describing a variety of settings. In the US, for example, these protective behaviors were associated with reduced incidence of SARS-CoV-2 in a large cohort study [120], with mask use retaining its effectiveness even in communities with limited physical distancing. In Bangladesh [25], the spread of SARS-CoV-2 was reduced in communities that were encouraged to wear facial masks. In Guinea-Bissau, however, distributing cloth masks did not help mitigate the impact of the pandemic on population health in local communities [121].

Our panel data also provide insights about how individuals adapt their behaviors when the threat from SARS-CoV-2 is receding, as was the case in October-November 2020 in Malawi. Whereas participants reported using masks in the past month frequently even at such a time of epidemic decline (round 4), the consistency of this behavior changed sharply when the recorded incidence of SARS-CoV-2 dropped in Malawi. The proportion of participants who

reported always wearing a mask in public was almost cut in half between August-September 2020 and October-November 2020. These changes in patterns of mask use occurred even though various places where people interact and congregate reopened (e.g., schools and universities), and individuals increasingly re-engaged in social ceremonies (e.g., weddings).

In our study, we found that the pace at which mask use was adopted varied significantly between population groups. Among study participants, early adopters of mask use often resided in urban areas, and had higher educational attainment than other respondents. Panel participants who occupied office jobs were also more likely to be early adopters, compared to those who worked in manual jobs, or those who engaged in business, retail and sales activities. These results highlight the need for targeted interventions to encourage mask use among specific socioeconomic groups, who might lag in their uptake of this intervention despite the health threat stemming from a new pandemic. Whereas community-wide strategies that included the free distribution of masks, the broad diffusion of information, and extensive engagement by community leaders, were successful in increasing mask use in several communities of Bangladesh [25], additional outreach might be needed in Malawi to reach members of groups with slower adoption patterns.

Women were also less likely than men to be early adopters of facial masks to prevent the spread of SARS-CoV-2. Even though their use of masks caught up over time, and they were not more likely than men to be classified as "laggards" in the adoption of this new health behavior, initial delays might have increased their exposure to the new health threat. Our study thus highlight the need for strategies to remedy gender-related inequality in access to, and the implementation of, preventive behaviors such as mask use [122]. This is particularly important in settings like Malawi, where vaccination coverage remains low.

Our study suffers from several limitations. First, respondents might have over-reported their use of protective measures such as mask use and physical distancing. In a study in Kenya, for example, self-reported mask use greatly exceeded levels of mask use documented by independent observers of public spaces [123]. In our study, incentives to report compliance with mask use recommendations might have increased after the use of masks in public spaces was mandated in August 2020, and fines were announced for non-compliers. Since this mandate coincided with the epidemic peak, this might have created a spurious temporal association between increasing reports of mask use and declining case counts or symptoms.

Second, our data on patterns of contact lack a proper baseline. Our pre-COVID study did not include questions on attendance of places and events, and the first round of the COVID-19 panel occurred in April/May, a few weeks after the Government had declared COVID-19 a national disaster, and subsequently closed schools and universities, and imposed limitations on the size of public gatherings. Reductions in attendance of some social events (e.g., weddings) thus likely occurred before our study baseline. These changes in contact patterns might have delayed the start of the outbreak and reduced its magnitude, however they did not fully prevent the local transmission of SARS-CoV-2 (Fig 2). Our data also do not cover subsequent waves of the COVID-19 pandemic in Malawi, which were characterized by the spread of more transmissible variants of the SARS-CoV-2 virus. The role of mask use and physical distancing in controlling those larger waves remains unclear.

Third, we only collected data on the consistency of mask use in rounds 3 (August-September) and 4 (October-November). We are thus unable to determine whether mandating the use of masks in August 2020 not only increased the proportion of individuals who wore masks for SARS-CoV-2 prevention, but also helped ensure consistent use among those who adopted this measure.

Fourth, there are sampling limitations. Despite high enrollment and follow-up rates, the sample size available for some analyses was small, particularly for urban areas. Due to limited

statistical power, we might not have been able to detect differences in trends in behaviors or attendance of places and events, by area of residence. Because it relied on mobile phones to conduct interviews, our sample was also selective [24]. Groups with higher levels of access to mobile phones were thus likely over-represented in our panel. Even though the study's sample was dispersed throughout the country, it was not representative of the population of Malawi.

Fifth, even though we used a flexible specification, our multivariate models either failed to identify covariates of the adoption of preventive behaviors, or only accounted for a small proportion of the variability in practices across panel participants and over time. This might have been the case because we focused on a small set of independent variables, which described key sociodemographic characteristics of respondents. As in studies of attitudes and behaviors related to other diseases (e.g., HIV), more detailed investigations of the adoption of preventive behaviors related to COVID-19 could explore factors related, for example, to an individual's social networks or his sources of information [124, 125]. Unfortunately, we did not collect such data in our panel study, in large part due to the need to keep interview duration short in surveys conducted by mobile phone.

Sixth, the contrast that we draw between Malawi and other African countries with regards to modalities of SARS-CoV-2 prevention may not be as stark as suggested by the documentation of official policies and measures. In particular, some countries that have issued stay-at-home or shelter-in-place orders might only have applied them partially, or might have faced significant resistance in their attempt to implement and enforce such orders.

Finally, our analyses of behavioral changes only suggest temporal associations with epidemic trends. Reductions in SARS-CoV-2 incidence documented by PCR testing (Fig 2) were preceded by, or occurred at approximately the same time as, the adoption of several protective behaviors in our sample (e.g., physical distancing, mask use). These associations cannot be interpreted as causal, because we cannot rule out competing explanations for this reversal of epidemic trends. For example, the decline in incidence/symptoms of SARS-CoV-2/COVID–19 might have been prompted by successes in contact tracing, or changes in environmental factors affecting the transmissibility of SARS-CoV-2 (e.g., temperature). Future investigations of the determinants of SARS-CoV-2 trends in Malawi and elsewhere should triangulate multiple sources of data (e.g., behavioral interviews, viral sequences, serosurveys, contact tracing datasets).

## Conclusion

Our study has described how individuals navigated the new health threat posed by COVID-19 in Malawi, a country where no requirements to stay at home were imposed during the first months of the pandemic. In that context, study participants reported wearing masks and practicing physical distancing as their main responses to the spread of SARS-CoV-2. The implementation of these strategies coincided with an epidemic downturn, suggesting that behaviors that reduce the transmissibility of SARS-CoV-2 during contacts might have played a key role in controlling the first wave of the COVID-19 pandemic in Malawi. The adoption of mask use and physical distancing was however uneven in our sample, with members of various disadvantaged socioeconomic groups reporting a delayed adoption of these protective practices. Our findings thus stress a) the importance of fostering protective behavioral changes to contain future outbreaks of SARS-CoV-2 in settings where access to vaccination remains low, and b) the need for targeted outreach to members of socioeconomic groups in which the adoption of protective behaviors might be delayed.

## Supporting information

**S1 Appendix. Geographic distribution of respondents.**
(DOCX)

**S2 Appendix. Changes in household occupancy ratio.**
(DOCX)

**S3 Appendix. Adoption of behaviors to reduce contacts between infected and susceptible individuals in Malawi (April to November 2020).**
(DOCX)

**S1 File. Replication files.**
(ZIP)

**S1 Checklist.**
(DOCX)

## Author Contributions

**Conceptualization:** Jethro Banda, Amelia C. Crampin, Georges Reniers, Abena S. Amoah, Stéphane Helleringer.

**Data curation:** Jethro Banda, Albert N. Dube, Amelia C. Crampin, Abena S. Amoah.

**Formal analysis:** Stéphane Helleringer.

**Funding acquisition:** Amelia C. Crampin, Georges Reniers, Stéphane Helleringer.

**Investigation:** Jethro Banda, Albert N. Dube, Abena S. Amoah.

**Methodology:** Georges Reniers, Stéphane Helleringer.

**Project administration:** Jethro Banda, Albert N. Dube, Sarah Brumfield, Amelia C. Crampin, Abena S. Amoah.

**Resources:** Abena S. Amoah.

**Software:** Jethro Banda, Sarah Brumfield.

**Supervision:** Jethro Banda, Albert N. Dube, Sarah Brumfield, Amelia C. Crampin, Abena S. Amoah.

**Validation:** Stéphane Helleringer.

**Visualization:** Stéphane Helleringer.

**Writing – original draft:** Jethro Banda, Stéphane Helleringer.

**Writing – review & editing:** Jethro Banda, Albert N. Dube, Sarah Brumfield, Amelia C. Crampin, Georges Reniers, Abena S. Amoah, Stéphane Helleringer.

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
