## [Decision Letter · Decision Letter 0]

12 Feb 2024

PGPH-D-23-02229

Controlling the first wave of the COVID–19 pandemic in Malawi: results from a multi-round study

Dear Dr. Helleringer,

Thank you for submitting your manuscript to PLOS Global Public Health. After careful consideration, we feel that it has merit but does not fully meet PLOS Global Public Health’s publication criteria as it currently stands. Therefore, we invite you to submit a revised version of the manuscript that addresses the points raised during the review process.

EDITOR:

I request you to kindly consider the reviewers comments to the best of your ability. The reviewers comments are forwarded to you as is. Kindly keep in mind that you may use your best judgement while responding to reviewers comments. In case of any concerns please do not hesitate to reach out to the editorial office.

We look forward to receiving your revised manuscript.

Kind regards,

Rashmi Josephine Rodrigues, M.D., Ph.D.

Academic Editor

Journal Requirements:

If you did not receive any funding for this study, please simply state: “The authors received no specific funding for this work.

3. Some material included in your submission may be copyrighted. According to PLOS’s copyright policy, authors who use figures or other material (e.g., graphics, clipart, maps) from another author or copyright holder must demonstrate or obtain permission to publish this material under the Creative Commons Attribution 4.0 International (CC BY 4.0) License used by PLOS journals. Please closely review the details of PLOS’s copyright requirements here: PLOS Licenses and Copyright. If you need to request permissions from a copyright holder, you may use PLOS's Copyright Content Permission form.

Potential Copyright Issues:

Fig 1: please (a) provide a direct link to the base layer of the map (i.e., the country or region border shape) and ensure this is also included in the figure legend; and (b) provide a link to the terms of use / license information for the base layer image or shapefile. We cannot publish proprietary or copyrighted maps (e.g. Google Maps, Mapquest) and the terms of use for your map base layer must be compatible with our CC-BY 4.0 license. 

Additional Editor Comments (if provided):

Reviewers' comments:

Reviewer's Responses to Questions

**Comments to the Author**

1. Does this manuscript meet PLOS Global Public Health’s publication criteria? Is the manuscript technically sound, and do the data support the conclusions? The manuscript must describe methodologically and ethically rigorous research with conclusions that are appropriately drawn based on the data presented.

Reviewer #1: Yes

Reviewer #2: Yes

Reviewer #3: Yes

2. Has the statistical analysis been performed appropriately and rigorously?

Reviewer #1: I don't know

Reviewer #2: Yes

Reviewer #3: Yes

3. Have the authors made all data underlying the findings in their manuscript fully available (please refer to the Data Availability Statement at the start of the manuscript PDF file)?

Reviewer #1: Yes

Reviewer #2: No

Reviewer #3: Yes

4. Is the manuscript presented in an intelligible fashion and written in standard English?

Reviewer #1: Yes

Reviewer #2: Yes

Reviewer #3: Yes

5. Review Comments to the Author

Reviewer #1: Is it possible to include a notion about governmental communication regarding the preventive behaviors studied? Where all measures pushed in a similar way in Malawi, or did you have politicians for example stressing the use of mouth masks and being an example already before they became compulsory?

Reviewer #2: Controlling the first wave of the COVID–19 pandemic in Malawi: results from a multiround Study

The topics of this paper are interesting, though well known in other contexts. The structure and content must be revised, and results have to be better explained by authors before to be reconsidered for publication.

Abstract has to be clarified the goal, provide empirical results, and health and social policy implications of results.

Introduction has to better clarify the research questions of this study and provide more theoretical background (See suggested readings that must be all read and used in the text).

Hypothesis is too long; it has to be shortened in 2-3 lines.

Section 2 has to "theoretical background" as title.

First, authors have to avoid a lot of subheadings that create fragmentation and confusion. If necessary, can use bullet points (same comments for all sections).

Authors have to better describe the different sources of transmission dynamics of COVID-19 (e.g., climate, air pollution, density, etc.) and risk factors in society, which can accelerate diffusion of novel coronavirus in environment and relate nonpharmaceutical measures of control. After that they can focus on the topics of this study to provide a correct analysis for fruitful discussion (See suggested readings that must be all read and used in the text).

Methods of this study is not clear. The section of Materials and methods must be re-structured with following three sections only and same order to be clear for readers:

• Sample and data

• Measures of variables. They have to be clearly indicated and described

• Models and Data analysis procedure.

Results.

To reiterate, avoiding sub-headings that create fragmentation of the paper.

Table 1 has to indicate in title that it is related to Malawi and the period under study.

Pseudio-R2 is low this suggest a misspecification of the model. I suggest revising the measures used in modelling.

Figure 2. Data have to be normalized with population on y-axis to be reliable. Analysis it should be better that is done also considering the third wave.

Figure 3. avoid in legend acronyms and use full words. The arrows are not clear what indicate. In any case this figure is not clear…and messy.

Figure 4 and 5-6 same comments for figure 3.

Analysis are related to 2020, and they have to better used to show how can be used to face next pandemics of new viral agents, when drugs are missing, in particular in terms of timing of mask wearing and behavioral change.

The paper has a lot of figures that are difficult to digest, some of them can be put in appendix and inserting in the text the most important ones to improve the readability…

As there is a comparison between rural and urban residents, a reliable comparative analysis should consider the differences of values to show the dynamics and design a strategy of intervention.

Clarify urban residents to which cities they belong.

Discussion.

First, authors have to synthesize the main results in a simple table to be clear for readers and then show what this study adds compared to other studies. Authors should provide with bullet points, clear best practices of health policy to cope with pandemic diffusion in rural and urban residents.

Conclusion has to be an autonomous section. Conclusion has not to be a summary, but authors have to focus on manifold limitations of this study and provide clear implications of health policy, as well as how nations can prevent and mange with good governance and new technologies (mechanical ventilation) next pandemics with vaccination and nonpharmaceutical measures of control.

Overall, then, the paper is interesting, but theoretical framework is weak, and some results are vague and create confusion… structure of the paper has to be improved; study design, discussion and presentation of results have to be clarified using suggested comments.

To be clear, I strongly suggest improving the paper, by using all comments (suggested papers included to read and use all) that I will in-depth verify, and maybe it can be considered. If the paper is not improved as suggested it will be dismissed.

Suggested readings of relevant papers that have to be read and all inserted in the text and references.

Chirwa, G.C., Zonda, J.M., Mosiwa, S.S., Mazalale, J. 2023. Effect of government intervention in relation to COVID-19 cases and deaths in Malawi.Humanities and Social Sciences Communications, 10(1), 335

Coccia M. 2023. High potential of technology to face new respiratory viruses: mechanical ventilation devices for effective healthcare to next pandemic emergencies, Technology in Society, vol. 73, May 2023, n. 102233, https://doi.org/10.1016/j.techsoc.2023.102233

Anscombe, C., Lissauer, S., Thole, H., ...Masesa, C., Gondwe, J. 2023. A comparison of four epidemic waves of COVID-19 in Malawi; an observational cohort study.BMC Infectious Diseases, 23(1), 79

Benati I.,Coccia M. 2022. Effective Contact Tracing System Minimizes COVID-19 Related Infections and Deaths: Policy Lessons to Reduce the Impact of Future Pandemic Diseases. Journal of Public Administration and Governance, vol. 12, n. 3, pp. 19-33. DOI: https://doi.org/10.5296/jpag.v12i3.19834

Banda, J., Dube, A.N., Brumfield, S., ...Crampin, A.C., Helleringer, S. 2021. Knowledge, risk perceptions, and behaviors related to the COVID-19 pandemic in Malawi.Demographic Research, 44, pp. 459–480

Coccia M. 2023. Sources, diffusion and prediction in COVID-19 pandemic: lessons learned to face next health emergency[J]. AIMS Public Health, 2023, 10(1): 145-168. doi: 10.3934/publichealth.2023012

Odenigbo, C., Crighton, E. 2023. Exploring COVID-19 from the perspectives of healthcare personnel in Malawi.Health Care Science, 2(4), pp. 242–254

Coccia M. 2022. Preparedness of countries to face COVID-19 pandemic crisis: Strategic positioning and underlying structural factors to support strategies of prevention of pandemic threats, Environmental Research, Volume 203, n. 111678, https://doi.org/10.1016/j.envres.2021.111678.

Ardito L., Coccia M., Messeni Petruzzelli A. 2021. Technological exaptation and crisis management: Evidence from COVID-19 outbreaks. R&D Management, vol. 51, n. 4, pp. 381-392, https://doi.org/10.1111/radm.12455

Tshotetsi, L., Hajison, P., Jella, C.D., Mpachika-Mfipa, F., Chimatiro, C.S. 2023. Knowledge, practices and adherence to COVID-19 preventive measures by community members in the Phalombe District Malawi: a cross-sectional qualitative study.Global Health Promotion

Núñez-Delgado A., Bontempi E., Coccia M., Kumar M., Farkas K., Domingo, J. L. 2021. SARS-CoV-2 and other pathogenic microorganisms in the environment, Environmental Research, Volume 201, n. 111606, https://doi.org/10.1016/j.envres.2021.111606.

Benati I., Coccia M. 2022. Global analysis of timely COVID-19 vaccinations: Improving governance to reinforce response policies for pandemic crises. International Journal of Health Governance. https://doi.org/10.1108/IJHG-07-2021-0072

Chawinga, W., Singini, W., Phuka, J., ...Sambani, C., Kambalame, D. 2023. Combating coronavirus disease (COVID-19) in rural areas of Malawi: Factors affecting the fight.African Journal of Primary Health Care and Family Medicine, 15(1), a3464

Bontempi E., Coccia M., 2021. International trade as critical parameter of COVID-19 spread that outclasses demographic, economic, environmental, and pollution factors, Environmental Research, vol. 201, Article number 111514, https://doi.org/10.1016/j.envres.2021.111514

Peters, M.A., Ahmed, T., Azais, V., ...Hansen, P.M., Shapira, G. 2023. Resilience of front-line facilities during COVID-19: evidence from cross-sectional rapid surveys in eight low- and middle-income countries.Health Policy and Planning, 38(7), pp. 789–798

Coccia M. 2021. The impact of first and second wave of the COVID-19 pandemic: comparative analysis to support control measures to cope with negative effects of future infectious diseases in society. Environmental Research, vol. 197, June, n. 111099, https://doi.org/10.1016/j.envres.2021.111099

Rosario Denes, K.A., Mutz Yhan, S., Bernardes Patricia, C., Conte-Junior Carlos, A., 2020. Relationship between COVID-19 and weather: case study in a tropical country. Int. J. Hyg Environ. Health 229, 113587.

Phiri, M.M., Macpherson, E.E., Panulo, M., ...Chunda, P., Morse, T. 2022. Preparedness for and impact of COVID-19 on primary health care delivery in urban and rural Malawi: a mixed methods study.BMJ Open, 12(6), e051125

Coccia M. (2020). Factors determining the diffusion of COVID-19 and suggested strategy to prevent future accelerated viral infectivity similar to COVID. The Science of the total environment, 729, 138474. https://doi.org/10.1016/j.scitotenv.2020.138474

Bontempi E., Coccia M., Vergalli S., Zanoletti A. 2021. Can commercial trade represent the main indicator of the COVID-19 diffusion due to human-to-human interactions? A comparative analysis between Italy, France, and Spain, Environmental Research, vol. 201, Article number 111529, https://doi.org/10.1016/j.envres.2021.111529

Thindwa, D., Jambo, K.C., Ojal, J., ...French, N., Flasche, S. 2022. Social mixing patterns relevant to infectious diseases spread by close contact in urban Blantyre, Malawi.Epidemics, 40, 100590

Coccia M. 2022. COVID-19 pandemic over 2020 (with lockdowns) and 2021 (with vaccinations): similar effects for seasonality and environmental factors. Environmental Research, Volume 208, 15 May 2022, n. 112711. https://doi.org/10.1016/j.envres.2022.112711

Kalina, M., Kwangulero, J., Ali, F., Tilley, E. 2022. You need to dispose of them somewhere safe: Covid-19, masks, and the pit latrine in Malawi and South Africa.PLoS ONE, 17(2 February), e0262741

Yuan, J., Li, M., Lv, G., Lu, Z.K. 2020. Monitoring transmissibility and mortality of COVID-19 in Europe ((2020) International Journal of Infectious Diseases, 95, pp. 311-315.

Flaxman, S., Mishra, S., Gandy, A., Unwin, H.J.T., Mellan, T.A., Coupland, H., Whittaker, C., (...), Bhatt, S. 2020. Estimating the effects of non-pharmaceutical interventions on COVID-19 in Europe (Open Access), (2020) Nature, 584 (7820), pp. 257-261.

Banda, J., Dube, A.N., Brumfield, S., ...Crampin, A.C., Helleringer, S. 2021. Knowledge, risk perceptions, and behaviors related to the COVID-19 pandemic in Malawi. Demographic Research, 44, pp. 459–480

Coccia M. 2021. How do low wind speeds and high levels of air pollution support the spread of COVID-19? Atmospheric Pollution Research, vol. 12, n.1, pp. 437-445., https://doi.org/10.1016/j.apr.2020.10.002.

Rahimi, N.R., Fouladi-Fard, R., Aali, R., (...), Conti Gea, O., Fiore, M. 2021 Bidirectional association between COVID-19 and the environment: A systematic review Environmental Research, 194,110692

Coccia M. 2021. Pandemic Prevention: Lessons from COVID-19. Encyclopedia, vol. 1, n. 2, pp. 433-444. doi: 10.3390/encyclopedia1020036

Obasa, A.E., Singh, S., Chivunze, E., ...Rennie, S., Moodley, K. 2020.Comparative strategic approaches to COVID-19 in Africa: Balancing public interest with civil liberties.South African Medical Journal, 110(9), pp. 858–863

Akan, A.P.; Coccia, M. 2023. Transmission of COVID-19 in cities with weather conditions of high air humidity: Lessons learned from Turkish Black Sea region to face next pandemic crisis, COVID, vol. 3, n. 11, 1648-1662, https://doi.org/ 10.3390/covid3110113,

Coccia M. 2021. High health expenditures and low exposure of population to air pollution as critical factors that can reduce fatality rate in COVID-19 pandemic crisis: a global analysis. Environmental Research, vol. 199, Article number 111339, https://doi.org/10.1016/j.envres.2021.111339

Li, H.-L., Yang, B.-Y., Wang, L.-J., (...), Ma, R.-F., Yang, X.-D. 2022. A meta-analysis result: Uneven influences of season, geo-spatial scale and latitude on relationship between meteorological factors and the COVID-19 transmission, Environmental Research 212,113297

Coccia M. 2023. Effects of strict containment policies on COVID-19 pandemic crisis: lessons to cope with next pandemic impacts. Environmental science and pollution research international, 30(1), 2020–2028. https://doi.org/10.1007/s11356-022-22024-w

Reviewer #3: This well-written paper is potentially useful to guide other researchers and practitioners in other COVID-19 studies. Saying that, I have a few comments, meaning to improve the discussion.

The findings confirm previous studies indicating the efficacy of mask-wearing, even in settings of poor social distancing in reducing COVID-19 transmission (https://www.nature.com/articles/s41467-021-24115-7). However, some caution is needed to interpret the results.

1) (a) Malawi´s age structure indicates that only 2.68% of the population was 65+ yo in 2020, making the elderly, a more vulnerable population, relatively small;

(b) The urban population accounts for only 18% of the total population - and COVID-19 transmission is much reduced in rural settings due to the sparser population.

So, it is no surprise that the number of severe cases and deaths was less than initially expected. In this situation, it should be interesting to discuss these two factors in this context.

2) Figure 1: We should view those map indicators with a grain of salt. The fact that the central governments issued a decree regarding stay-at-home orders does not mean that those orders were effectively applied. This is especially important in countries where the presence of the state is precarious and lacks the structure to enforce the lockdowns and provide adequate health care.

3) Figure 6: According to the graphs, rural and urban residents show nearly identical behavior regarding mask use, contrary to what was stated in some parts of the text.

4) It should be interesting to discuss the paper's findings in light of the following statement

"Closing all educational institutions, limiting gatherings to 10 people or less, and closing face-to-face businesses each reduced transmission considerably. The additional effect of stay-at-home orders was comparatively small",

extracted from the article

https://pubmed.ncbi.nlm.nih.gov/33323424/

6. PLOS authors have the option to publish the peer review history of their article (what does this mean?). If published, this will include your full peer review and any attached files.

**Do you want your identity to be public for this peer review?** For information about this choice, including consent withdrawal, please see our Privacy Policy.

Reviewer #1: No

Reviewer #2: No

Reviewer #3: **Yes: **Luiz H. Duczmal

---

## [Decision Letter · Decision Letter 1]

12 Aug 2024

PGPH-D-23-02229R1

Controlling the first wave of the COVID–19 pandemic in Malawi: results from a multi-round study

Dear Dr. Helleringer,

Thank you for submitting your manuscript to PLOS Global Public Health. After careful consideration, we feel that it has merit but does not fully meet PLOS Global Public Health’s publication criteria as it currently stands. Therefore, we invite you to submit a revised version of the manuscript that addresses the points raised during the review process.

Reviewer 3 has provided a minor concern, could you please ensure that you address this final request?

We look forward to receiving your revised manuscript.

Kind regards,

Johanna Pruller, Ph.D.

Staff Editor

Journal Requirements:

Additional Editor Comments (if provided):

Reviewers' comments:

Reviewer's Responses to Questions

**Comments to the Author**

1. If the authors have adequately addressed your comments raised in a previous round of review and you feel that this manuscript is now acceptable for publication, you may indicate that here to bypass the “Comments to the Author” section, enter your conflict of interest statement in the “Confidential to Editor” section, and submit your "Accept" recommendation.

Reviewer #2: All comments have been addressed

Reviewer #3: All comments have been addressed

2. Does this manuscript meet PLOS Global Public Health’s publication criteria? Is the manuscript technically sound, and do the data support the conclusions? The manuscript must describe methodologically and ethically rigorous research with conclusions that are appropriately drawn based on the data presented.

Reviewer #2: Yes

Reviewer #3: Yes

3. Has the statistical analysis been performed appropriately and rigorously?

Reviewer #2: Yes

Reviewer #3: Yes

4. Have the authors made all data underlying the findings in their manuscript fully available (please refer to the Data Availability Statement at the start of the manuscript PDF file)?

Reviewer #2: Yes

Reviewer #3: Yes

5. Is the manuscript presented in an intelligible fashion and written in standard English?

Reviewer #2: Yes

Reviewer #3: Yes

6. Review Comments to the Author

Reviewer #2: I have read thoroughly the revised version of paper.

Now this version of the paper after revision done is OK and provides interesting results for readers.

Reviewer #3: I´m satisfied with the authors´response to my comments.

Just a minor comment, which could be left at the discretion of the authors:

1) Please fix Figure 1: the Hala'ib Triangle region on the Egypt-Sudan border is missing from the map.

7. PLOS authors have the option to publish the peer review history of their article (what does this mean?). If published, this will include your full peer review and any attached files.

**Do you want your identity to be public for this peer review?** For information about this choice, including consent withdrawal, please see our Privacy Policy.

Reviewer #2: No

Reviewer #3: **Yes: **Luiz Duczmal

---

## [Editor Report · Decision Letter 2]

20 Sep 2024

Controlling the first wave of the COVID–19 pandemic in Malawi: results from a multi-round study

PGPH-D-23-02229R2

Dear Helleringer,

We are pleased to inform you that your manuscript 'Controlling the first wave of the COVID–19 pandemic in Malawi: results from a multi-round study' has been provisionally accepted for publication in PLOS Global Public Health.

Best regards,

Julia Robinson

Executive Editor